# Audeo: Audio Generation for a Silent Performance Video

**Kun Su**[*]    **Xiulong Liu**[*]    **Eli Shlizerman**[†*]

## Abstract

We present a novel system that gets as an input, video frames of a musician playing the piano, and generates the music for that video. The generation of music from visual cues is a challenging problem and it is not clear whether it is an attainable goal at all. Our main aim in this work is to explore the plausibility of such a transformation and to identify cues and components able to carry the association of sounds with visual events. To achieve the transformation we built a full pipeline named '*Audeo*' containing three components. We first translate the video frames of the keyboard and the musician hand movements into raw mechanical musical symbolic representation Piano-Roll (Roll) for each video frame which represents the keys pressed at each time step. We then adapt the Roll to be amenable for audio synthesis by including temporal correlations. This step turns out to be critical for meaningful audio generation. In the last step, we implement Midi synthesizers to generate realistic music. *Audeo* converts video to audio smoothly and clearly with only a few setup constraints. We evaluate *Audeo* on piano performance videos collected from YouTube and obtain that their generated music is of reasonable audio quality and can be successfully recognized with high precision by popular music identification software. The source code with examples is available in a Github repository [3].

## 1  Introduction

> *Melody is the essence of music. I compare a good melodist to a fine racer.*
> *Wolfagang Amadeus Mozart*

The perfect combination of a musician's skills with the musical instrument tones creates the delightful experience of 'live music'. Such an event is inspiring from the perspective of the melody being played and also from the perspective of witnessing admirable synchrony between the musician and the instrument.

What makes the musical performance to sound as it sounds? The answer to this question is intertwined. We know many of the components that make musical performance sound well, but what we do not know is how to rigorously quantify the contribution of the components. Notes, tempo, consistency, timed precision, mechanical accurateness, rhythmic movements, harmonics, frequencies; all these and more delicately compose the melody of a musical piece. Quantifying these aspects plays a key role in the attempt to better understand how to generate realistic melodies.

A particular test which informs regarding music generation is to constitute the music (transcribe the music) from visual information, i.e., finding possible ways to recreate the audio stream of a musical performance just from the visual stream. In the case of a piano recording, that would be

---

[*]Department of Electrical & Computer Engineering, University of Washington, Seattle, USA
[†]Department of Applied Mathematics, University of Washington, Seattle, USA

[3]https://github.com/shlizee/Audeo

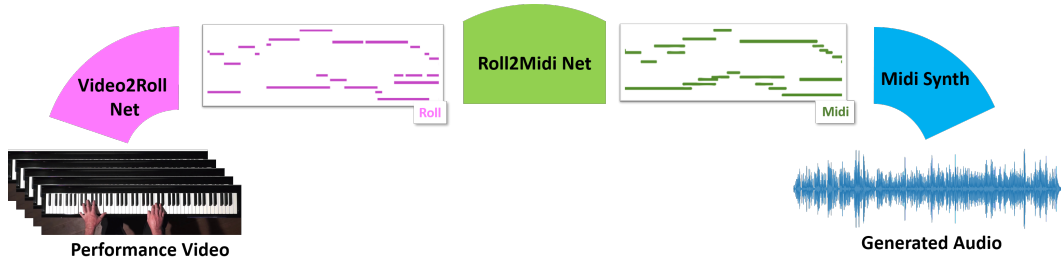

Figure 1: Given an input of video frames of musician playing the piano, *Audeo* generates the music for that video. Please also see supplementary video and materials with sample results.

taking into account the positions of the musician's hands, body, the positions of the keys and the pedals and merge them into music. Timed precision between visual cues and sounds is known to have a profound effect on such a task and takes the form of a far more complicated process than a mere synchronization. The reasons for the complexity stem from visual stream perception being of significantly slower rate than the perception of an audio stream, however, the perception of their combination requires the latency between the audio and the video signals to be faster than the rate of the visual stream. This creates an effect in which for the generation of an audio signal for a video, one should not only find an association between the video frames and the audio but also to precisely complete the audio stream in between the video frames going back and forth between the past and the future frames. Such completion is nontrivial and requires exhaustive knowledge of the instrument and body mechanics, i.e., a model of a virtual instrument, or an ability to imagine the details from the visual features, similar to a composer's ability to envision the melody from reading musical notes.

Video frames include an abundance of visual information, some of which could be irrelevant to music. Therefore, it is plausible that instead of a direct transformation, intermediate features could be used for the translation from video to audio. These features should capture the mechanical and the perceptual features of the interaction between the musician and the instrument and be constructive tools for sound representation and synthesis. For example, the Musical Instrument Digital Interface (Midi) protocol is a candidate signal. It is used to interchange musical information between instruments and encodes various keyboard functions and musical attributes. Variants of Midi, such as Pseudo-Midi (binary, without expressive velocities), will provide an even more compact version to encode keyboard function and musical attributes altogether. Moreover, connecting visual actions with frequencies of the audio signal as it varies with time, i.e., the Spectrogram, can be a useful mediator.

In this work, we address the challenge of music generation from video by proposing a full pipeline, named *Audeo*, to generate the audio of a silent piano performance video. *Audeo* translates the performance from the video domain to the audio domain in three stages, through the recovery of mediator signals. In the first stage, given a top-view video, we use multi-scale feature attention deep residual network to capture the visual information and to predict which keys are pressed at each frame (Video2Roll Net). We formulate this as a multi-label classification task, and the collection of predictions can be seen as a 'Piano-Roll' [1]. However, 'Roll' is still coarse binary prediction and does not directly correspond to Pseudo-Midi critical for music synthesis. Therefore, in the second stage, we utilize a Generative Adversarial Network (GAN)[2] to refine and enhance the Roll with musical attributes to output the Pseudo-Midi signal (Roll2Midi Net). This step turns out to be critical for providing symbolic musical representation. The third and last stage of the *Audeo* pipeline is the synthesis of Pseudo-Midi to the audio signal (Midi Synth). Since the predicted Pseudo-Midi is binary and missing expressive velocities, we thereby propose to use the same velocity to synthesize a mechanical audio via a classical Midi synthesizer, or a deep synthesizer to obtain more realistic audio. The deep synthesizer translates Pseudo-Midi to a spectrogram and then to audio. An overview of the *Audeo* system is shown in Fig. 1. Our main contributions are the following: (i) To the best of our knowledge, we are the first work to transcribe the music audio from silent piano performance videos that are not recorded in specific lab setting. (ii) We introduce a full pipeline, named *Audeo*, containing three interpretable components to complete this transformation. (iii) *Audeo* is robust and generalizable and we show that the output audio of piano performances that *Audeo* generates will be consistently detected by popular music identification software as the expected musical piece.

## 2 Related Work

While audio-visual signals are interrelated, classically, there has been a clear separation of these signals into a single domain of video or audio. Earlier traditional methods based on canonical correlation analysis (CCA) [3] or a likelihood criterion [4] were used to localize and track audio-associated visual objects. Deep learning approaches have succeeded in connecting the two streams and began the consideration of joint audio-visual tasks. Systems have been proposed to leverage and explore the correlation of both audio and video simultaneously, i.e., *audio-visual cross-domain tasks*. For example, conditioning the visual and sound streams on each other as training supervision was shown as an effective training method for networks with unlabeled data in audio-visual correspondence [5, 6, 7, 8]. Moreover, it was shown that it is possible to separate object sounds by inspecting the video cues of an unlabeled video [9, 10, 11, 12] or to perform audio-visual event localization task on unconstrained videos [13] and even to generate natural sounds, e.g., baby crying, water flowing, given a visual scene [14]. The latter generation task is conceptually similar to the task that we consider, however, it is on a much slower scale than the generation of music and the visual input in the case of the piano playing.

Each direction of audio and video relation has been studied as well. In the *audio-to-video* direction, deep learning RNN based strategies were proposed to generate body dynamics correlated with sounds from audio-only [15, 16, 17]. Moreover, systems that generate parts of the face or synchronize lips movements from speech audio were shown to be possible [18, 19]. In the *video-to-audio* direction, prior work addressed the identification of objects which are most correlated with sounds. For piano performance, that would be the keyboard, musician's hands, etc. Combinations of traditional computer vision techniques were presented to provide these functionalities [20, 21, 22]. However, these methods turned out to be sensitive to the environment setup, such as the camera position, illumination condition, and so on. In order to improve performance, the use of depth cameras was proposed to detect the pressed keys with depth information, however, while it indeed improved accuracy, such a strategy cannot be generalized to unconstrained videos [23, 24]. Furthermore, machine learning methods such as Support Vector Machine (SVM) [25] were proposed to classify a single key status, whether it is pressed or not [26]. Since these methods required large datasets of manual labels to be trained on, systems using deep learning methods, such as Convolution Neural Networks (CNN), have been applied to key identification problem as well, approaching the problem as a binary classification task where each single key needs to be cropped separately and labeled manually before training and testing [27, 28]. Deep learning strategies also addressed both audio and video streams to identify actions such as estimation of following musical notes, i.e., a two-stream CNN has been proposed to determine the notes being played at any moment for the task of identifying whether correct fingers and keys are used for the corresponding notes [29]. Very recently, Foley Music [30] utilizes body keypoints from silent videos to synthesize plausible music.

The methods described above require training sets with an associated *Ground Truth Midi*. Such Midi is typically obtained with an electronic keyboard, a process that makes the creation of the training data to be limited. To overcome this challenge, the Onsets and Frames framework enables to transcribe audio waveform to Midi [31]. Recent work used this framework to obtain Pseudo Ground truth Midi and implemented a ResNet [32] to predict the pitch onsets events (times and identities of keys being that have been pressed) given video frames stream [33]. While this method achieves an acceptable prediction of onsets, there is still a gap between the onset prediction problem and the reconstruction of a complete Midi containing the offsets as well. The combination of onsets and offsets would provide the ability to generate music. *Audeo* is using the Onsets and Frames framework to obtain a Pseudo Ground Truth Midi for training as well and thus can be applied to any video for top view. Moreover, *Audeo* generalizes the prediction task and generates a complete and robust Midi for synthesis via either traditional or deep learning-based Midi synthesizers.

In music generation, several deep learning approaches have been introduced. Autoregressive models that directly work on audio waveform such as Wavenet [34], SampleRNN [35] and their variants [36, 37] have shown successes in both speech and music generation. However, the transformation between two different domains (e.g., text to speech (TTS), the symbolic musical score to audio) is more challenging. Tacotron [38, 39] proposed the encoder-decoder architecture to translate text to Mel-spectrogram and a Wavenet conditioned on generated Mel-spectrogram to generate final human speech waveform. In addition, Timbretron [40] uses CycleGAN [41] for timbre style transform on spectrogram level. Recently, non-autoregressive models like MelGAN [42] also

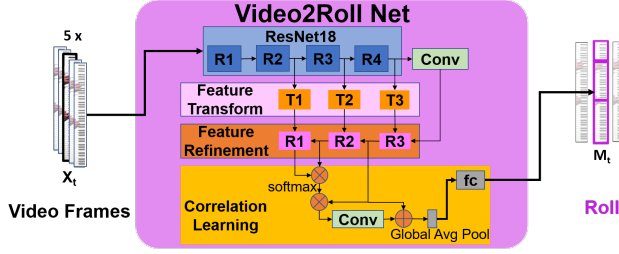

Figure 2: Detailed schematics of the components in VIDEO2Roll Net: ResNet18 + feature transform, feature refinement and correlation learning. Input: 5 consecutive frames; Output: pressed-key prediction at the middle frame.

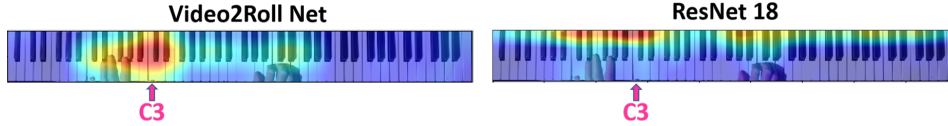

Figure 3: Visualized feature maps comparison between Video2Roll Net (left) and ResNet18 (right) using Scored Weighted Class Activation Heatmap (Score-CAM) [45]. This example demonstrates that our method can locate the delicate visual cues of pressed C3 key more accurately.

demonstrated convincing results on audio generation. However, unlike common human speech which is monophonic, piano music is challenging to generate due to its polyphonic property. In addition, symbolic Midi can be viewed as a time-frequency representation (while the text transcript for speech cannot). Since music is polyphonic and contains more content information, the TTS model cannot be directly applied to score-to-audio generation. While a conditional Wavenet has been proposed to enable Midi synthesis [43], training a conditional Wavenet requires exhaustive computation resources. Another efficient possibility is to use Performance-Net (PerfNet) [44] which has been shown to successfully and efficiently convert Midi to the spectrogram. The last step of *Audeo* uses the pre-trained PerfNet as a deep-learning-based Midi synthesizer to generate audio in the spectrogram domain.

## 3 Methods

Our key approach is to use generalizable and interpretable mediator signals to translate piano video frames to output audio. Indeed, for piano performance, these are videos from the Internet (usually without accompanied Ground Truth Midi). We retrieve the Pseudo Ground Truth (GT) Midi from the audio with the Onset and Frames framework [31]. This allows us to avoid hardware constraints of the instrument and to use any video, even those recorded in an unconstrained setup. The Pseudo GT Midi can be considered as a two-dimensional binary matrix $M \in \mathbb{R}^{K \times T}$ where $K$ is the number of pitches and $T$ is the number of frames. For each entry, $M_{k,j}$, 1 indicates if the key $k$ is sustained at frame $j$ and 0 otherwise. We describe the details of each component of the *Audeo* system in the following subsections.

**Video2Roll Net:** The task in this stage can be defined as a multi-label image classification problem. One video clip $X$ can be seen as a four-dimensional tensor $X \in \mathbb{R}^{T \times C \times H \times W}$ where $T$, $C$, $H$, $W$ are time, channel, height and width dimension respectively. We use stacked five consecutive grayscale frames $X_{t-2,t-1,t,t+1,t+2}$ as the input into Video2Roll Net which outputs a prediction of the keys pressed in the middle frame $X_t$. Mathematically, we estimate the conditional probability of the keys being pressed at frame $t$ given video frames $X_{t-2:t+2}$. The probability of estimated keys at frame $t$ will be $P(\hat{M}_{:,t}) = P(M_{:,t}|X_{t-2,t-1,t,t+1,t+2})$. We find that the use of consecutive frames is critical to detect changes in the pressed keys. Note that estimating all pressed keys at each frame is a harder task compared to the prediction of onsets events only (which and when a key is being pressed) as considered in [33]. We use ResNet18 as the backbone, similar to [33], but our architecture takes into consideration the natural phenomena appearing in this task: 1) the visual cues of the sustained keys are relatively small compared to other objects in the image such as hands and fingers;

2) at each frame, the pressed keys may correlate due to the concept of musical harmony so some combinations have a higher chance to appear at the same time than others; 3) the spatial dependencies are significant to detect the sustained keys but the typical CNN is designed to be invariant to spatial positioning. To address these issues, we design a multi-scale feature attention network similar to [46]. Specifically, using ResNet18 as the backbone, Video2Roll Net contains three functional modules: feature transform, feature refinement, and correlation learning. The feature refinement setup is similar to a feature pyramid network (FPN) [47] which uses top-down features propagation mechanism. The main difference to common FPN is that in our Video2Roll Net multi-scale features at residual blocks are first transformed and re-calibrated via feature transform module before passing to the next stage. This allows the network to detect the visual cues on various scales better. As a final component, the correlation learning module is used to learn feature spatial dependencies and semantic relevance by the self-attention mechanism. The response of any location to attention is related to the features of other locations. We use the refined output features $R1$ and $R2$ to compute the attention weight matrix with the semantic relevance of features considered. Compared with features $R1$ and $R2$, $R3$ has rich semantic information. Therefore, we use the learned attention matrix to regularize $R3$. Since detection of pressed keys is essential to generate meaningful music, the multi-scale feature attention strategy enables Video2Roll Net to find the region of visual cues associated with pressed keys more accurately (as shown in Fig. 3).

**Roll2Midi Net:** The prediction of the Piano Roll (Roll) $\hat{M}_{:,1:T}$ of Video2Roll Net is not perfect due to various challenges. For example, hand occlusions in video frames pertain Video2Roll Net from detecting changes in pressed keys. Moreover, because $\hat{M}_{:,t}$ is predicted at each frame individually, Roll predictions do not have a temporal correlation. Also, since Pseudo GT Midi is generated from the Onset and Frames framework [31], which depends on the audio stream, one common phenomenon is that if the performer sustains a key for a sufficiently long time, the magnitude of the corresponding frequency will gradually decay to zero and this key in the Pseudo GT Midi will be marked as off, however, since our Video2Roll Net depends on short-time visual information only, all pressed keys are still considered as active. Hence this prediction will not match the reality of the audio. Examples can be seen in Fig. 6. In both black and green frames, Video2Roll Net detects more active keys than in Pseudo GT Midi since these keys are indeed pressed in the frames but marked as inactive in the Pseudo GT Midi. We call this effect a mismatch of audio-visual information. To mitigate these effects, we introduce a generative adversarial network (GAN) [2] to refine and complete the Video2Roll results $\hat{M}_{:,1:T}$ so that the outputs are closer to Pseudo GT Midi. The GAN includes a generator $G$ and a discriminator $D$. The input of the generator is Roll predictions $\hat{M}_{:,T_1:T_2}$ and each column of $\hat{M}$ is the probability score retrieved from the last fully connected layer of Video2Roll Net after applying a sigmoid function. Using probability scores instead of threshold outputs enables the generator to re-calibrate the probabilities and to generate a more robust Pseudo Midi representation. The GAN objective is defined by:

$$\min_{G} \max_{D} \mathbb{E}_{M \sim \mathcal{M}}[\log D(M)] + \mathbb{E}_{\hat{M} \sim \hat{\mathcal{M}}}[log(1 - D(G(\hat{M})))]. \tag{1}$$

Our generator is a five depths U-Net [48] and the discriminator consists of 5 layers CNN. Having a discriminator instead of simply applying a U-Net, allows the model to learn a more general pattern of the Pseudo Midi and the prediction becomes acceptable once it is 'real' enough. Since the variations of Midi in different music styles are significant, using U-Net only may overfit to the training data. We use the Mean Square Error (MSE) to optimize both the generator and the discriminator. During inference, we pass the Roll representation to the generator and obtain the refined representation (Midi) $\hat{M}_R = G(\hat{M})$. The Roll2Midi Net can boost the correctness of overall predictions and the estimated Midi is sufficient to be synthesized to get meaningful music close to the ground truth. Fig. 5 shows that Roll2Midi can partially eliminate the false positives and false negatives in the Roll.

**Midi Synth:** Both the Roll and the Pseudo Midi can be synthesized to audio using classical Midi synthesizers. We find that it is sufficient to get clear, robust, and reasonable music with the predicted Midi. Moreover, the classical Midi synthesizer is flexible and can support creative applications. For example, music with various timbres can be generated using piano performance video only by simply setting instruments other than piano during the synthesis step. While interesting results can be obtained at this point, the audio synthesized from classical Midi synthesizers is mechanical since the predicted Pseudo Midi is binary and does not include the expressive velocities. Moreover, estimating expressive velocities on the Midi-level requires to have a Midi GT which specifies them, however the Onsets-and-Frames that we use as the GT generates velocity prediction with insufficient precision. We

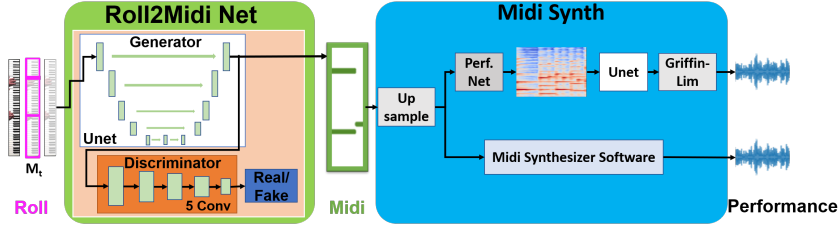

Figure 4: Detail schematic of Roll2Midi Net and Midi Synth components of *Audeo* system.

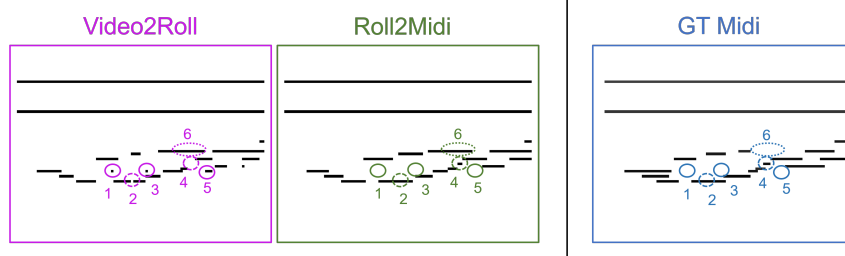

Figure 5: Comparison of Roll, Midi and Pseudo GT Midi. Solid ellipses (1,3,5) : elimination of false positives; Dashed ellipses (2,4): elimination of false negatives; Dotted ellipse (6): failure not eliminated.

thereby investigate whether we can generate more realistic music with the Pseudo Midi predictions via deep synthesizers. To do that, we pre-train a PerfNet [44] with Pseudo GT Midi $M$. The PerfNet learns a transformation $H$ between $M$ and the spectrogram $S$. With the pre-trained PerfNet, we forward propagate the Midi $\hat{M}_R$ to obtain an initial estimated spectrogram $\hat{S}_R = H(\hat{M}_R)$. Note that even though our predicted Pseudo Midi has been refined, a discrepancy between $M$ and $\hat{M}_R$ would still exist and we find that using PerfNet to learn transformation from $\hat{M}_R$ to $S$ directly can't be generalized. We conjecture that this is due to the sensitivity of the transformation between the Pseudo Midi and the spectrogram which increases the difficulty in the generalization. To mitigate this problem, we train an additional U-Net to do the refinement on the spectrogram level. This U-Net can be formulated as a function $U$ and we aim to minimize the L1 distance between $\hat{S}_R$ and $S$: $L_1(\hat{S}_R, S) = \|U(\hat{S}_R) - S\|$. We find that estimating the initial rough spectrogram first and then performing the refinement on the spectrogram level later leads to better generalization. As the last step, Griffin-Lim algorithm is used to convert the spectrogram to an audio waveform [49].

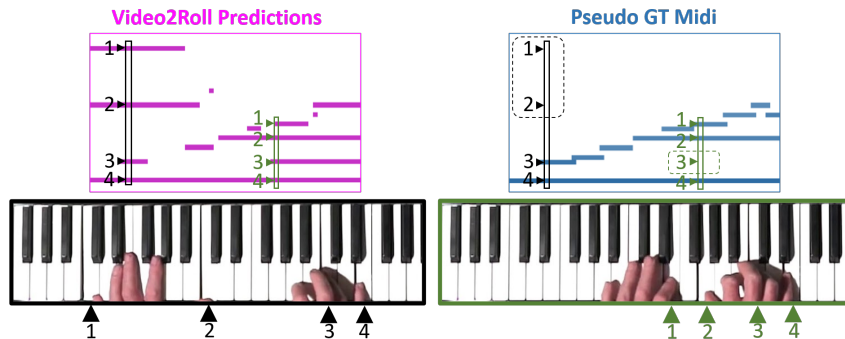

Figure 6: Examples of Pseudo GT Midi mismatches with pressed keys. Keys that are active in our predictions and in video frames (black and green) are marked as off in Pseudo GT Midi (dashed).

# 4  Experiments & Results

**Datasets.** In contrast to previous works that were designed or tested in a specific lab setting, we evaluate *Audeo* pipeline directly on piano performance videos available on YouTube. The minor constraint for data collection is top view piano performance with a fully visible keyboard. Indeed, the instrument and the camera setup are not required to be the same for the recordings that we use. Particularly, we use videos recorded by Paul Barton[4] at the frame rate of 25fps and the audio sampling rate of 16kHz. The Pseudo GT Midi are obtained via Onsets and Frames framework (OF) [31]. Since OF has a low precision on expressive velocity prediction, all Pseudo GT Midi sets are set as binary and are down-sampled to 25fps. We crop all videos and keep the full keyboard only and remove all frames that do not contribute to the piano performance (e.g. logos, black screens, etc). We trim the initial silent sections up to the first frame in which the first key is being pressed, to align the video, Pseudo GT Midi, and the audio. All silent frames inside each performance are kept. Two evaluation sets are used in our experiments.

*Pseudo Midi Evaluation Set:* This set is used to evaluate our predictions in Video2Roll Net and Roll2Midi Net. We use 24 videos of Bach Well-Tempered Clavier Book One (WTC B1) which includes a total of 115 minutes of training data. The testing set contains the first 3 Prelude and Fugue performances of Bach Well-Tempered Clavier Book Two (WTC B2) which in total includes 12.5 minutes. This results in $172,404$ training images and $18,788$ testing images. The Pseudo GT Midi data that we are using is not perfect, as the GT Midi, and may indeed impair evaluation. We thereby include additional audio evaluation protocols described below.

*Audio Evaluation Set:* This set is for audio evaluation only. We aim to test whether the generated music can be detected by music identification software. This test set contains 35 videos from WTC B2 (24 Prelude and Fugue pairs and their 11 variants), 8 videos from WTC B1 variants, and 9 videos from other composers. This combination results in 52 videos and 297 minutes in total.

**Evaluation Metrics.** For the *Midi Evaluation*, we evaluate predictions from our Video2Roll Net and Roll2Midi Net by reporting the precision, the recall, the accuracy, and the F1 score on the frame-level defined in [50]. To compare with other methods, we reproduce proposed models in [33] and test them on our Pseudo Midi Evaluation set. For *Audio Evaluation*, we use the popular music identification App SoundHound[5] to perform detection test on the generated music. We split every performance into multiple 20 second segments and perform the detection test on every segment once. The detection is marked as a success if SoundHound successfully shows the correct source name of the music and failure if nothing or the wrong source shows up. We report the average detected rate at the segment level. Furthermore, we evaluate the results with a Human Perceptual test using Amazon Mechanical Turk (see Suppl. Materials).

## 4.1  Implementation details

**Video2Roll Net:** We implement the elimination of data biases such as color, piano shapes, by setting all image frames to grayscale, a crop of keyboard region, and transformation to common frame size ($100 \times 900$). Due to the imbalanced label classes in the Midi evaluation set, we force each training mini-batch to contain classes evenly by over/downsampling strategy. Features obtained at residual blocks are used to do feature-level transform and refinement except for the first block. We train the network using binary cross-entropy loss with a batch size of $64$.

**Roll2Midi Net:** We extract probability scores (without threshold) from Video2Roll Net at each frame and concatenate them as the Roll representation. We use a 4 seconds Roll (100 frames) during training. The five depth U-net generator takes one channel input and each depth and down-samples the height and width by half. The discriminator includes five convolution layers that take the Pseudo Midi as input and classify it as real or fake. Both the generator and the discriminator are trained with MSE loss with the batch size of $64$.

**Midi Synth:** We use FluidSynth[51] as classical Midi synthesizer. For all results, we set the initial tempo to be $80$ and velocity for all active keys be $100$. For the deep synthesizer, a PerfNet is pre-trained with Pseudo GT Midi using MSE loss with a batch size of $16$. The target spectrogram of an audio clip is the magnitude part of its short-time Fourier Transform. We compute the log-scaled spectrogram with $2,048$ window size and $256$ hop size, leading to a $1025 \times 126$ spectrogram for 2 seconds audio sampled at 16kHz. The 2 seconds Pseudo Midi (50 frames) are up-sampled to 126

| Model | Precision | Recall | Accuracy | F1-score |
|---|---|---|---|---|
| ResNet [33] | 64.3 | 54.7 | 40.4 | 49.7 |
| ResNet+Aggregation+slope [33] | 61.5 | 57.3 | 41.2 | 50.8 |
| Video2Roll Net (Our) | 61.2 | 65.6 | 46.4 | 56.4 |
| Roll2Midi Net TS=0.4 (Our) | 60.0 | **77.0** | 50.6 | **61.5** |
| Roll2Midi Net TS=0.5 (Our) | **65.1** | 69.9 | **50.8** | 60.4 |

Table 1: Precision, recall, accuracy and F1-score in (%) for pseudo Midi evaluation. If not specified, all results use threshold (TS) = 0.4 after the application of the sigmoid function. Bold number indicates the best result.

| | Total | Bach WTC B1 Variants | Bach WTC B2 & Other |
|---|---|---|---|
| ResNet+FluidSynth | 55.9 | 74.2 | 52.9 |
| Roll+FluidSynth | 62.6 | 79.6 | 59.6 |
| Midi+PerfNet | 73.0 | 80.6 | 71.6 |
| Midi+FluidSynth | **73.9** | **85.6** | **72.4** |
| Ground Truth | 89.2 | 92.6 | 87.7 |

Table 2: Sound Hound music identification rate in (%).

frames to fit with the input shape size of PerfNet. Once we obtain the initial spectrograms from PerfNet, we train a five depths Unet to refine the spectrogram. Since the highest frequency on a piano key is $4186.01$ Hz, we use for training only the frequency bins up to $576$. As the last step, we use the Griffin-Lim algorithm [49] to generate the final audio.

All networks in our *Audeo* system are trained in PyTorch [52] using the Adam optimizer [53] with $\beta_1 = 0.9$, $\beta_2 = 0.999$. For all models, we use the learning rate starting from $0.001$ and gradually decreasing it if the validation loss is in a plateau. Two Nvidia Titan X GPUs are used to train all components in *Audeo*. More specific implementation details can be found in the Supplementary Materials.

## 4.2 Results

**Midi Evaluation:** Table 1 shows the results of *Audeo* on generation of the Roll and the Pseudo Midi compared to other methods. The Video2Roll Net detects detailed visual cues that result in higher recall, accuracy, and F1-score compared to previous works. It turns out that having fewer false negatives is essential to generate a complete melody without missing the notes. The relatively low precision of Video2Roll Net reflects the fact that mismatches in audio-visual information are a common phenomenon (See Fig. 6). Thereby, we do expect false positives in the predictions. Indeed, music generation from visual information is nontrivial and this is one of the common challenges and we believe that it will be enhanced in the future. The results indicate that to get cleaner and robust symbolic representation Roll2Midi Net is indeed necessary. The core of the generative adversarial network enables Roll2Midi Net to partially eliminate both false negatives and false positives by judging whether the generated Pseudo Midi is real enough. Indeed, Roll2Midi Net boosts the overall performance even further. The F1 score of Roll2Midi outperforms the best model in [33] by more than $10\%$.

**Audio Evaluation on Music Identification:** We compare the detection by SoundHound of samples generated from *Audeo* system to ResNet baseline and the ground truth audio. Furthermore, we synthesize Roll and Pseudo Midi obtained from *Audeo* via FluidSynth or PerfNet to test and exclude synthesizer effects. The results of music identification are shown in Table 2. Note that the Bach WTC B1 has already been learned during training and we use their variants to evaluate whether *Audeo* is robust to different performance styles such as fast tempo, staccato, legato, and so on. It turns out that all *Audeo* methods outperform the ResNet baseline and synthesizing Pseudo Midi via FluidSynth or PerfNet can reach more than $80\%$ detected rate while Midi+FluidSynth achieves the best accuracy ($85.6\%$). This is compared to the Ground Truth detection of ($92.6\%$). This indicates that *Audeo* can capture the core of learned music and is not sensitive to variance in performance. For test videos from the type that was not introduced in training at all, such as Scott Joplin, both Midi+PerfNet and Midi+FluidSynth pass a $70\%$ detected rate while ResNet baseline obtains $52.9\%$. While the gap with the ground truth ($72.4$ vs $87.7\%$) is still obvious, the identification results demonstrate the robustness and generality of the *Audeo* system. In terms of total average, Midi+FluidSynth performs better

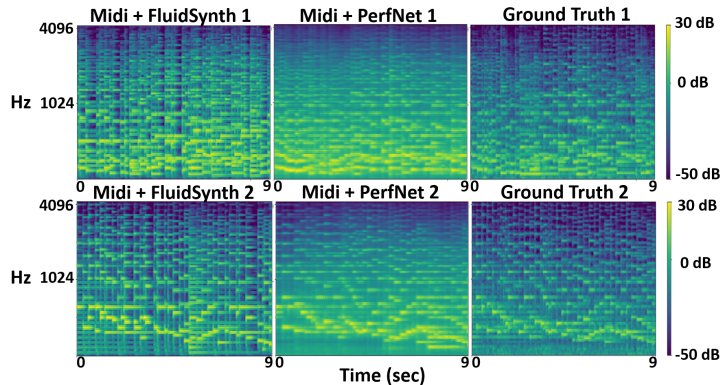

Figure 7: Two samples of generated spectrograms vs Ground Truth.

than other methods and outperforms ResNet baseline by $18\%$. Notably, using PerfNet as synthesizer results in slightly lower detection than FluidSynth in this test. While deep synthesizer may recover emotion and naturalness in the spectrogram domain, it also introduces noise which is non-trivial to reduce. Fig.7 compares spectrograms of samples synthesized via FluidSynth and PerfNet. While both syntheses produce similar spectrograms to the ground truth one, we observe that Midi+FluidSynth is cleaner but having the same velocity for all notes results in an unnatural sound. On the other hand, Midi+PerfNet has magnitude variance and changes smoothly in time but noticeable noise exists.

## 5 Possible Future Applications and Conclusion

An important guideline for our pipeline was interpretability and modularity of implementation such that it would be possible to incorporate it in various video-music applications with piano. An immediate type of application would be on-the-fly music transform (e.g., as we show a timbre transform here). Adding a camera on top of a general piano keyboard (no need for electronic) could generate various timbres and can be possibly implemented in real-time applications. An extension of such a real-time application would be a virtual piano, where in a virtual-reality environment, without the need for a mechanical instrument at all, the pipeline could produce a full virtual piano experience. We also foresee applications that analyze video-audio streams in a post-processing manner. For example, by mounting a camera on top of the piano it could be possible to isolate the piano transcription from a multi-instrument performance, without affecting the performance, or, *Audeo* pipeline may be combined with current audio-only piano transcription methods. Indeed, as we discuss here, the additional visual cues detected and processed by the pipeline could be matched for audio-visual synchrony and enhance the output.

In conclusion, we present a novel full pipeline system, named *Audeo*, for generating music from silent piano performance video. Each component in *Audeo* is an interpretable component and flexible to be used for various practical purposes, such as key detection, piano learning synchronization, timbre modulation, etc. Experimental results demonstrate that *Audeo* can effectively generate reasonable music that can be detected by music identification software.

## Acknowledgement

We acknowledge the support of Washington Research Foundation Innovation Fund. The authors would also like to acknowledge the partial support by the Departments of Electrical & Computer Engineering (KS and ES), Applied Mathematics (ES), the Center of Computational Neuroscience (ES), and the eScience Center (ES) at the University of Washington in conducting this research.

## Broad Impact

The *Audeo* system enables music generation from silent piano performance video. One classical application is to recover a corrupted audio channel in a piano performance video. Moreover, since *Audeo* uses Midi as an intermediate representation, this provides a large amount of possibilities to manipulate the generated Midi creatively. For example, people could use the predicted Midi to synthesize music of any instrument by just giving a piano performance video. This can be also extended to virtual piano environment in the real-time where *Audeo* can generate music from visual information when there is no real sound available at all. All these directions would benefit from *Audeo*. Due to the fact that the generated music can be detected by a music identification App, one concern could be the possibility that a fake pianist could scam audiences utilizing *Audeo* system. This is a common concern in the application of any generative model. Failure in *Audeo* may bring up unsatisfying music but we do not expect serious consequences. Also, while *Audeo* is trained and tested on videos of the same pianist, we believe the full pipeline is valid and robust in general due to the variance in the pianist can only result in the difference in hand shape and this variance can easily be incorporated in *Audeo* by either fine tuning or adding more data to the training set.

## Footnotes

[4] https://www.youtube.com/user/PaulBartonPiano

[5] https://www.soundhound.com/

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
