[Supplementary Material]

# Supplementary Material
## NeurIPS 2020 submission ID 2811

## 1   Generated Audio Samples and Overview Video

Please see the attached video for a short overview and generated audio samples. Please turn Audio ON.

## 2   Perceptual Evaluation of the Generated Music

In addition to the evaluation of the different components of *Audeo* with music recognition software (Soundhound) presented in the main paper, we also performed human perceptual survey using Amazon Mechanical Turk. We performed the survey in order to provide an additional source of evaluation to the recognition test. In the survey we asked people (non-experts) to listen to the Ground Truth (GT) first and then to listen and choose a single version of generated music from video corresponding to the GT that is the most similar to the GT. The given options of generated music were: Roll+FluidSynth, Midi+FluidSynth and Midi+PerfNet. No background on the survey or the *Audeo* project was given to the participants to avoid any perceptual biases. We surveyed 50 participants individually, where each participant was asked to evaluate 10 random segments of GT music each with twenty seconds *(500 segments in total)* along with three generated versions.

Figure 1: Human evaluation result

The results are shown in Fig. 1. We observe a clear indication that Midi generated segments (89% chose

these options) are more similar to the GT than Roll generated segments (only 11% chose this option) and correspond to the Soundhound evaluation as well. Between Midi+PerfNet and Midi+FluidSynth we almost an equal split between their similarity to the GT with a slight preference for FluidSynth synthesizer, which is also consistent with Soundhound recognition evaluation. In particular, out of 89%, 46% associate Midi+FluidSynth with the GT while 43% associate Midi+PerfNet with the GT.

In summary, the perceptual survey indicates that the recognition metric that we use to evaluate the generated music is a robust metric correlated with the perception of the music. The metric thus could have a potential to be useful to further refine the generation and to analyse the sources of inconsistencies.

# 3 Additional Details of the Implementation

## 3.1 Video2Roll Net

We provide additional details on the feature transform and feature refinement blocks of Video2Roll Net. The overall structure of these blocks was proposed in [1] and we describe our implementation of them in Video2Roll Net.

**The Feature Transform Block:** We use the feature transform block to transform different level features into the same low-dimension space (see Fig. 2). The input feature is first passed to a convolutional layer with $1 \times 1$ kernel size to reduce the channel size to 128. Then two additional convolutional layers with $3 \times 3$ kernel size are performed. In between these two layers, batch normalization and Relu activation are applied. As a last step, a residual connection is used to get the transformed feature.

Figure 2: The components and the flow within the Feature Transform block.

**The Feature Refinement Block:** Following the feature transform block which transforms features at different levels into the same low dimensional space, we implement feature refinement block to fuse and recalibrate these features (see Fig. 3). Specifically, we first concatenate the high- and low-level features along the channel dimension. Then global average pooling is applied to capture the global context information. It outputs a vector $z \in \mathbf{R}^{C \times 1 \times 1}$ where $C$ is the channel size. We set $C = 128$ in our implementation. This vector is then passed to two fully connected layers with Relu activation in between, and followed by a sigmoid activation which structures the output to be a gate-weighted vector. We expand this weighted vector to have same spatial dimension as of the input low-level feature dimension and perform an element-wise multiplication to obtain the refined low-level feature.

Figure 3: The components and the flow within the Feature Refinement block.

## 3.2  Roll2Midi Net

**Generator:** The generator structure is a U-net with 5 depth levels (also discussed in the main paper). In the direction of down-sampling, we use a convolutional layer with $3 \times 3$ kernel size followed by a batch normalization and a leaky Relu activation function. The number of channels increases from 1 to $64, 128, 256, 512, 1024$ with propagating downward with each level. The up-sampling is done by a transposed convolution with $3 \times 3$ kernel followed by a batch normalization and a leaky Relu activation function where the corresponding depth features in down-sampling direction are concatenated together. The number of channels decreases from 1024 to $512, 256, 128, 64, 16$ with propagating upward with each level. At the last step, a convolutional layer with $1 \times 1$ kernel size is used to get the 1 channel output to be of same dimensions as the input.

**Discriminator:** The discriminator contains five convolutional layers. The first four layers use $3 \times 3$ kernel size and increase the number of channels from 1 to $64, 128, 256, 512$. We set the stride as 2 for the first three layers and 1 for the last layer. At the last step, a convolution layer with $1 \times 1$ kernel size with stride 3 is used to bring the number of output channels back to 1 in order to determine whether it is real or fake.

# References

[1] Zheng Yan, Weiwei Liu, Shiping Wen, and Yin Yang. Multi-label image classification by feature attention network. *IEEE Access*, 7:98005–98013, 2019.