[Reviews · NeurIPS 2020]

Review 1

Summary and Contributions: This paper describes a method for automatically transcribing and synthesizing piano performances based on visual cues (i.e., an overhead view of the hands and keyboard). The proposed method first learns to generate a piano-roll representation from the video stream, and then synthesize the resulting piano roll via either spectrogram reconstruction and phase recovery, or using a traditional software midi synthesizer (fluidsynth). The proposed method is evaluated on a collection of piano performances gathered from youtube, and compares favorably to simplified methods along multiple criteria (transcription error, song identification). --- Update post discussion: I have revised my score for this paper up a notch. I still have some reservations about the work as noted in my initial review, but my overall impression is favorable.

Strengths: Although the methods used here are relatively standard at this point, the authors do a good job of systematically evaluating the different components (video->piano roll, piano roll-> midi, midi->synthesized audio).

Weaknesses: On the technical side, the main weakness that I see is that the data used for evaluation is quite narrow in scope. The authors claim that the method should generalize (line 318), but no supporting evidence for that claim is presented. I'm also a bit skeptical of using onsets-and-frames output as "(pseudo) ground truth" with no reported correctness checks. At a higher level, I'm struggling to understand the motivation for this work. The problem setting is such that the input to the model would require a rather specialized setup (top-down, overhead view of keyboard) that really only exists in tutorial videos. It's hard to imagine a scenario in which this is available but the audio is not.

Correctness: No theoretical claims are made, but the empirical methodology appears to be sound. It would be nice to see a little more detail about how transcription accuracy is computed -- using mir_eval? -- as these things can be quite subtle to get right.

Clarity: There are some strange grammatical quirks, but overall the paper is easy enough to follow.

Relation to Prior Work: Coverage of prior work seems sufficient to me.

Reproducibility: Yes

Additional Feedback: Outside the use-cases claimed by the authors, one of the motivations that I could see for using video to supplement a piano transcription (as opposed to just audio) is to help inform subtle properties of the music, like micro-timing or dynamics. The quantization of the midi data (binary activation, no velocity, 25Hz) might be hurting here though, as it may be too coarse to support that kind of evaluation. I understand that this work might be too early to get at those questions, but it seems like a promising direction to pursue. The choice of piano is also interesting, but the framing of the problem (literally, on the keyboard) makes it impossible to infer pedaling and sustain. I don't have any strong suggestions here, but it is something the authors should consider if they continue pursuing visual instrument transcription.


Review 2

Summary and Contributions: This paper introduces a model for generating piano music performances in MIDI format from a video. It has a nice application for isolating piano transcriptions in multi-instrument performance settings, which could be useful for musicians and music students. Though they are not the first to approach this problem, they go further than previous work (which only predicted onset times) by predicting both note onset and offset times. This enables audio synthesis via MIDI samplers.

Strengths: This paper introduces two models that improve on previous work from Koepke et al., which introduced the problem of visual piano transcription. The modeling choices in both proposed models (Video2Roll) and (Roll2Midi) are well motivated and demonstrate quantitative improvements. The paper is well written and the main concerns are raised and well addressed.

Weaknesses: In terms of application, this work seems like a somewhat incremental improvement over the visual piano transcription model from Koepke et al. I think the motivations for the audio synthesis portion of the work are not as well justified as for the MIDI transcription application.

Correctness: Yes, the methodology and claims look good.

Clarity: The paper is nicely written and clear to understand.

Relation to Prior Work: Yes

Reproducibility: Yes

Additional Feedback: Nice paper overall. I'd like to see a bit more of a detailed discussion that dives into the applicability of this method in practical situations. One of the main motivations (stemming from the Koepke et al. paper) is that this would allow transcription of the piano player even during an ensemble performance. But how often do concerts get recorded with a video from this angle on the pianist's hands? Will this method generalize beyond your test set? This seems like an important direction for this line of work going forward. -- Update Thanks for your responses to the reviews. One point to clarify from your response - "Even with the Midi, synthesis of music is not straightforward, since Midi is binary and missing expressive velocities" - the MIDI specification actually does include velocity, so this language (which I realize now is also reflected in the paper) is misleading. I understand your point, which is that by predicting onset times only, you only get timing but not velocity, but this is worth fixing. In the long term, I'm not convinced that your audio synthesis approach is the best way to go about the task of transcribing the velocities, but it's interesting that you explored this. I look forward to seeing the final version of the paper.


Review 3

Summary and Contributions: The paper proposes a method for directly generating audio for a silent video (top view of piano keyboard). This is done by: 1) Video2Roll: Predict the Piano-Roll from input video, more precisely which keys are pressed when. This is cast as a multi-label classification problem. The idea is the same as [27] but the network architecture is different (same backbone ResNet18 but then some additional modules for top-down information propagation and some self-attention) 2) Roll2Midi: Roll produced in (1) is noisy and has multiple flaws, so a GAN is used to predict the Midi representation. 3) Midi Synth: Midi is converted to audio by predicting the spectrogram (either via a classical Midi synthesizer or by training a network [38]) and then converting spectrogram to audio. All stages use automatically produced supervision as the 'ground truth' Midi is obtained automatically via [25].

Strengths: 1) First method to go directly generate music from input silent video. 2) Everything that is proposed is reasonable and makes sense. 3) Results look good. On Midi prediction there are comparisons between [27], the proposed Video2Roll, and Video2Roll + Roll2Midi (final) and the final method works convincingly better than the rest. Evaluation is also performed on the resulting audio by passing it through an external music recognition software which shows that the performance is quite good (of course it is a hard task so nobody expects it to be perfect).

Weaknesses: 1) Multiple design choices are not justified enough, there are many potential alternatives - why were these picked, were others tried, etc? For example: 1.1) Why exactly this form of the Video2Roll network, e.g. how to pick where the self-attention is happening? 1.2) Why use a GAN for Roll2Midi? One could just directly train to predict the ground truth Midi from the Video2Roll output? 1.3) The PerfNet for Midi to Spectrogram is trained with a reconstruction loss. So why not GAN given a GAN is used for Roll2Midi? 2) Only a single performer is ever used, in training and in testing. I understand the task is hard but it would be great if at least a few different performers were chosen in order to e.g. not overfit to the particular piano, appearance of the performer's hands, exact camera setup, performer's style of pressing keys, etc. I also find it a bit of an overstatement that music is generated 'in the wild' given that all videos come from the same performer, same camera setup (I think?) cropped to just show the keyboard etc.

Correctness: All good.

Clarity: In general the paper is clear, but a few points are not: 1) I understand space is limited but there's too little information about network architectures in the main paper (though most is addressed in supplementary). Not talking about details of filter sizes etc (ok to put in supplementary) but core ideas behind architectures are missing, e.g. "correlation learning" is not explained at all and we just have to figure it out from figure 2. 2) The explanation of the GAN is not clear - equation (1) just gives the objective for the fake/real distinction, but presumably the generator is also stimulated to produce the ground truth (otherwise it could just produce any music), which is not written here. I'm not sure what the MSE comment in L.188 means, maybe it addresses my point, i.e. generator is trained to produce the ground truth. But the comment states that the discriminator is also trained with MSE, what does this mean? Is it that instead of log loss authors use square loss? So then equation (1) is wrong? 3) L.217: Is the cropping and cleaning up done manually or automatically? Will you release this clean data? 4) Maybe I'm missing something, but it doesn't seem straight forward to balance the mini-batch (L.241) - this is easy for single-label cases but how to do it for multi-label? Weighting seems easier?

Relation to Prior Work: Overall yes, though I think some pre-deep learning citations would also be nice as audio-visual learning existed before that as well. E.g.: "Pixels that sound" Kidron et al. CVPR 2005 "Harmony in motion” Barzelay and Schechner CVPR 2007

Reproducibility: Yes

Additional Feedback: No need to respond, just please clarify these points in the next version: - What is the meaning of bold in table 1? - L.176: presumably the problem is not because 'it depends on visual information only', but because the input is short. If the input was long, even if it was visual only, the network could learn to turn off long presses. So the explanation is slightly wrong. - L.144 Confusing to call it a frame when it has a temporal dimension (5 frames) - L.246: Not sure what 'slide' means, is it meant to be 'stride'? But I don't see why a stride is being called a 'window' then? - L.41: an->and, L.42 whether -> either --------- Post rebuttal After reading other reviews I realize that indeed the motivation or potential applications of this type of work are somewhat lacking. There are some related tasks (e.g. isolating piano in multiinstrument performances) but for most of them audio is likely to also be available. I also agree with other reviewers (also in my original review) that "in the wild" is an overstatement given the limited variability (same performer, similar camera poses) of the train and test data. However, despite these problems I still think the paper should be accepted - the proposed method is solid, the task is quite challenging (though perhaps artificially overchallenging) justifying somewhat the relatively simple (wrt in the wild) data, and the results look good. I reduce my score from 8 to 7 accordingly.


Review 4

Summary and Contributions: This paper proposes a novel pipeline approach for improving piano music/audio generation from silent videos with a top-view of a pianist's fingers playing on a keyboard. Prior work [27] used an end-to-end approach to directly predict a symbolic piano performance from video using ResNets. This paper points out there's a lot of mismatch between the video and music/audio streams and hence the processing requires multiple stages of transformation. The proposed pipeline consists of three interpretable components / stages. 1) Video2Roll "transcribes" local visual movements in the video (middle frame of 5 frames, ~0.2 seconds) onto a binary roll as a multi-class classification problem. Video2Roll consists of three stages. 2) Roll2Midi uses a GAN to transform longer chunks of rolls (4 seconds, giving more musical context) predicted from the last stage to match pseudo ground-truth piano rolls transcribed from audio (transcribed using the onsets and frames model [25]). 3) The MIDI synthesis stage which uses a pre-trained PerfNet (trained on pseudo ground-truth piano rolls to expressive spectrograms) to synthesis spectrograms and then a U-net to refine the spectrograms to account for the mismatches in the predicted rolls and the pseudo ground-truth piano rolls used to train PerfNet, and finally the Griffin-Lim alogirithm is used to convert the spectrograms to audio. The experiments show that the proposed method (first two stages) improves drastically over prior work [27] on predicting the pseudo ground-truth piano rolls. The audio synthesis was evaluated by if the synthesized audio could be recognized by a popular music recognition software SoundHound as being a certain piece, also improving over prior work [27].

Strengths: The paper proposes a novel pipeline that aligns well with the challenges of the domain, improving over prior methods by looking closely at the failure modes (Figure 5 and 6) and addressing them. Visualizations such as Figure 3 are helpful in furthering our understanding in how each component enhances the performance (perhaps the thumb is a bit cut of from the piano and appears more in surrounding frames?). The included video demo is compelling. It also shows even though the models were trained on J.S. Bach's prelude and fugues for well-tempered claviers, they were able to generalize to a wide-range of composers. The paper offers insight into overcoming mismatch in distributions that often occur in tasks involving multiple modalities.

Weaknesses: The paper claims to work on "in the wild" piano performance videos but the empirical evaluations focus on one performer and one top-view setup. While prior work involves modeling components that extract the keyboard from the rest of the video and unrolls fish-eye effects to be robust to different top-view setups, this paper focuses on improving the generated music quality. The motivation for the task could be discussed more and also how it might strengthen other related tasks. The paper aims to capture the expressive velocity in the piano audio synthesis yet the audio synthesis results are not the strongest components of the paper. Might the modeling of expressive velocity be incorporated in an earlier stage of the pipeline, for example Roll2Midi through GANs? Could velocity be partially transcribed through video? The included supplementary samples show that the samples do well on pitch while the rhythm is at times uneven. What are some inherent challenges? Could the symbolic models be strengthened by pre-training on a wider range of piano music?

Correctness: The experiments in Table 1 show how each of the added components improve the results, similarly Table 2 shows how "recognizable" the generated music/audio is with each added component, while also considering generalization on performance styles. Using music recognition software such as Soundhound as a proxy to "recognizable" is interesting. Perhaps some discussion on if human evaluation may or may not add to the evaluation could be helpful.

Clarity: The paper is well-written. The prose seems to emphasize "melody" in the introduction and at various points of the paper yet the work models polyphonic music.

Relation to Prior Work: Are there related work in other cross-modality work that address the mismatch in distributions that might strengthen the context of this paper?

Reproducibility: No

Additional Feedback: Minor suggestions: - In the figures, the y-axis feels up-side down because lower pitches are higher on the y-axis. - In Table 2, it might be clearer to label “Midi” as Roll2Midi to distinguish from Midi? - Perhaps some discussion on pedaling. ---- post author feedback ---- Thank you to the authors for providing additional motivation, context, and clarifications. It could be helpful to see more comparison to related work in the technical approaches to addressing challenges in mismatch in distributions in cross-modality work. Regardless, the paper presents a thoughtful approach to a challenging application and hence I maintain my overall positive rating.

[Author Response · NeurIPS 2020]

We thank the reviewers for their positive view of our work and feedback. We address and clarify the items raised by the reviewers below. We will make sure to include the novel explanations and corrections in the camera ready version.

**Novelty & Motivation**: Transcription of piano performance video to music is a challenging task. State-of-the-art methods succeeded so far to predict coarsely onsets of notes, however, as we show, it is insufficient for realistically sounding music of the performance. With this work we thereby explore if it is *at all possible* to directly transcribe the music from video. Indeed, our work is the first full pipeline of transcribing (rather than generating similar) music with each component specifically designed to serve this purpose. Video2Roll is designed to capture detailed visual cues from video and transform them to binary prediction. We introduce components such as feature transform, feature refinement, and correlation learning which enhance Roll prediction. However, it is still a coarse binary prediction and does not directly correspond to pseudo-GT Midi (Figs. 5,6) critical for music synthesis. Our solution is to introduce a Roll to Midi context-conditional GAN, where given Roll, a Midi is generated to fool the discriminator. Even with the Midi, synthesis of music is not straightforward, since Midi is binary and missing expressive velocities. We thereby propose to use the same velocity to synthesize a mechanical audio, or PerfNet, a recent one-to-one deterministic transcription from Midi to Spectrogram. While the methods that we compose into the first two pipeline components, such as multi-scale features, self-attention mechanism, conditional GAN, Unet, have been generically introduced, they have not been systematically combined to transcribe music and as far as we know in any video-audio system. Our work thus identifies such necessary components (out of many examined) to experimentally succeed in music transcription.

**Possible Applications**: An important guideline for our pipeline was interpretability and modularity of implementation such that it would be possible to incorporate it in various video-music applications with piano. An immediate type of application would be on-the-fly music transform (e.g., we show a timbre transform in the paper). Adding a camera on top of a general piano keyboard (no need for electronic) could generate various timbres and can be possibly implemented in real-time application. An extension of such a real-time application would be a virtual piano, where in a virtual-reality environment, without need for mechanical instrument at all, the pipeline could produce a full virtual piano experience. In addition, we foresee applications that analyze video-audio streams in postprocessing manner. For example, as R2 suggests, with mounting a camera on top of the piano it could be possible to isolate the piano transcription from a multi-instrument performance, without affecting the performance, or as R1 suggests, our pipeline may be combined with current audio-only piano transcription methods. Indeed, as we discussed in the paper, the additional visual cues detected and processed by the pipeline could be matched for audio-visual synchrony and enhance the output.

**Generality**: In contrast to many previous works designed or tested in a specific lab setting, we aim to use general top-view Youtube videos not recorded for the purpose of our method with no specifics on the recordings settings. Indeed, the instrument and camera setup are *not required to be the same* for the recordings that we used. The performer played music on different pianos and aspect ratio of the keyboard under the camera was variable. During the pre-processing step, we implement elimination of biases such as color, piano shapes, by setting all frames to gray scale, crop of keyboard region, and transformation to common frame size ($100 \times 900$). One unavoidable bias is of the performer/s hands. However, since Video2Roll network is designed to focus on generic visual cues and not on the hand specifics, we expect that the importance of hands would not be significant when additional performers are in the dataset. Notably, we addressed other aspects of generalization such as variation in music style. We made sure that when training is limited to a single composer (e.g. Bach) our pipeline is tested on a variety of music styles and would transcribe music with similar quality as of composer used for training (e.g., see supplementary video *NeurIPS2020_2811_sup_video.mp4*). Our experiments suggest that additional videos would not necessarily improve precision. We foresee significant possible enhancement if the dataset is curated to have balances in terms of variations of keys to be detected.

**Response to R1**: Please see clarifications regarding motivation, possible applications and generalizations above. The Onsets-and-Frames(OF) framework has high average frame-level precision ($88.5\%$ without velocity prediction) on MAPS dataset. OF allows us to use videos from YouTube. We only use the binary representation, since OF has a low precision on expressive velocity prediction ($35.52\%$). The imperfect pseudo GT may indeed impair evaluation when done on Midi only and not tightly related to the transcribed music. We thereby include additional audio evaluation protocols (SoundHound and Human evaluation) (see line 231 in paper and supp. materials). As an initial work, we do not consider pedaling since it would be necessary to include an annotated video-audio dataset to explore this feature.

**Response to R2**: Please see clarifications regarding applications and generalizations above. We will elaborate further on the audio synthesis portion which deals with the problem that pseudo Midi (binary & missing velocities) synthesis to music is not straightforward. We propose to use the same velocity to synthesize a mechanical audio or to learn the expressive velocities implicitly via NN-based synthesizer (PerfNet).

**Response to R3**: We added the citations to Kidron et. al., 05' and Barzelay et. al., 07' on classical methods and plan to discuss them in the Related Work section of the camera ready version. We will also add details on system choice and fix the errors mentioned in the feedback.

**Response to R4**: Estimating expressive velocities requires to have a Midi GT which specifies them, however the Onsets-and-Frames(OF) that we use as GT generates velocity prediction with $35.52\%$ precision. We therefore propose instead to learn it implicitly via a NN-based synthesizer (PerfNet). Similarly for rhythm and pedaling, these would require high enough quality annotated GT such that they would add to enhanced audio output beyond PerfNet.

[Meta-Review · NeurIPS 2020]

All reviewers recommend acceptance. Their main concerns were about empirical evaluation, and the overall motivation and potentially limited scope of the work. I share the latter concern, but I do not believe that it should be grounds for rejection, given the quality of the work. A remaining concern is the quality of the training data, since labels are produced by an existing algorithm (onsets-and-frames) which is not 100% accurate. Please make sure to adequately address the implications of this in the updated manuscript.